# Deep-Learning-Based Approach for Automated Detection of Irregular Walking Surfaces for Walkability Assessment with Wearable Sensor

**Hui R. Ng** [1,*] , **Xin Zhong** [1] , **Yunwoo Nam** [2] and **Jong-Hoon Youn** [1,*]

1   Department of Computer Science, University of Nebraska Omaha, Omaha, NE 68182, USA; xzhong@unomaha.edu
2   Community and Regional Planning, University of Nebraska-Lincoln, Lincoln, NE 68588, USA; ynam2@unl.edu
*   Correspondence: hng@unomaha.edu (H.R.N.); jyoun@unomaha.edu (J.-H.Y.)

**Abstract:** A neighborhood's walkability is associated with public health, economic and environmental benefits. The state of the walking surface on sidewalks is a key factor in assessing walkability, as it promotes pedestrian movement and exercise. Yet, conventional practices for assessing sidewalks are labor-intensive and rely on subject-matter experts, rendering them subjective, inefficient and ineffective. Wearable sensors can be utilized to address these limitations. This study proposes a novel classification method that employs a long short-term memory (LSTM) network to analyze gait data gathered from a single wearable accelerometer to automatically identify irregular walking surfaces. Three different input modalities—raw acceleration data, single-stride and multi-stride hand-crafted accelerometer-based gait features—were explored and their effects on the classification performance of the proposed method were compared and analyzed. To verify the effectiveness of the proposed approach, we compared the performance of the LSTM models to the traditional baseline support vector machine (SVM) machine learning method presented in our previous study. The results from the experiment demonstrated the effectiveness of the proposed framework, thereby validating its feasibility. Both LSTM networks trained with single-stride and multi-stride gait feature modalities outperformed the baseline SVM model. The LSTM network trained with multi-stride gait features achieved the highest average AUC of 83%. The classification performance of the LSTM model trained with single-stride gait features further improved to an AUC of 88% with post-processing, making it the most effective model. The proposed classification framework serves as an unbiased, user-oriented tool for conducting sidewalk surface condition assessments.

**Keywords:** deep learning; gait analysis; sidewalk surface assessment; walkability; wearable sensor

## 1. Introduction

Walking is one of the most common forms of physical activity that promotes individual mental and physical well-being. There are many health, environmental and economic benefits associated with high walking levels in communities [1–4], which has led to increased interest in maintaining walkable neighborhoods through walkability assessments. Walkability can be defined as the way individuals perceive the quality of the walking environment by measuring the friendliness of the built environment to walking [1,4]. Although many built environment features are used to define walkability, the presence and quality of sidewalks serve as substantial indicators influencing the perceived safety and overall satisfaction within the pedestrian environment [5,6]. Therefore, sidewalk assessments are an essential component of walkability assessment tools [7].

Pedestrian interviews or surveys are examples of commonly practiced sidewalk assessment methods that consider pedestrians' perceptions [8]. Nevertheless, these responses can be biased and lack expert insight. Government agencies also rely on trained experts to

perform on-site inspections to identify violations of pre-defined regulations [9]. These conventional practices are ineffective and inefficient because they require significant financial expenditure, time and intensive labor.

Hitherto, the limitations mentioned above can be addressed with the advancement of sensing technology. The methods for automated sidewalk assessments can be divided into two categories: approaches based on infrastructure-based data collected from the surroundings and approaches based on wearable sensor data.

For approaches based on infrastructure-based data, various advanced methods have been presented for applying deep learning techniques on infrastructure-based data to automate sidewalk assessments. Some examples of these urban data include street-view images, videos or geographic information system (GIS) data. Several works have also explored using deep learning approaches on vehicle responses measured with smartphone inertial sensors mounted on vehicles to identify sidewalk and roadway anomalies. However, these approaches do not consider the involvement of pedestrians in their evaluations.

On the other hand, utilizing wearable sensors for automated sidewalk assessments considers individuals' behaviors. Wearable sensors can be placed on pedestrians to measure human physiological reactions and the signals captured can in turn be used to examine how human physiological reactions are influenced as a result of their surrounding environments. Earlier studies demonstrated that various sidewalk features or defects led to changes in human response. Yet, these studies focused on investigating the association of signal magnitude or specific gait parameters with surface conditions or built environments with statistical modeling.

Thus far, there have been limited studies involving pedestrians in sidewalk surface condition assessments using machine learning or deep learning techniques. Nevertheless, existing works were either not generalizable to individual differences due to the absence of gait features, thereby lacking the ability to capture biomechanical characteristics, or the focus was not on assessing sidewalk surface conditions for the purpose of a walkability assessment. To the best of our knowledge, none of the existing works explored incorporating gait analysis techniques with machine learning or deep learning approaches for sidewalk surface assessments. To bridge this gap, in our previous study [10], we identified the ideal body location for placing a sensor, which is at the right ankle. Then, we developed a traditional machine-learning-based classification framework for identifying irregular walking surfaces using gait features extracted from the right-ankle sensor to train machine learning models and demonstrated the effectiveness of the approach [10]. Nonetheless, the traditional machine-learning-based method requires manual feature extraction and selection, which can be time-consuming and labor-intensive. Considering one of the main advantages of deep learning is its automated feature learning capability on raw data [11], adopting a deep-learning-based approach could circumvent manual feature extraction and selection. Furthermore, when a deep-learning-based approach is fed with handcrafted features, knowledge can be distilled from those features to improve performance [12]. Therefore, in this paper, we introduced a novel deep-learning-based classification framework that analyzed the acceleration data from a single right-ankle sensor for the automated detection of irregular walking surfaces. The acceleration data collected from the right-ankle sensors of 12 subjects from our previous experiment were labelled and applied to our proposed approach. Concretely, because the acceleration data were collected in a time series manner, we proposed using a deep LSTM network. Three different input modalities—raw accelerometer data, single-stride and multi-stride hand-crafted accelerometer-based gait features—were explored, and their effects on the classification performance of the proposed framework were compared and analyzed. To verify the effectiveness and feasibility of our approach, the classification performance of our proposed approach was compared to the traditional machine-learning-based method from our previous study.

This paper provided the following contributions:

1.  We presented a novel classification framework using a deep LSTM network to distinguish good and irregular walking surfaces with a single wearable sensor placed on the right ankle. To the best of our knowledge, no existing studies in the walkability assessment domain have proposed a framework that combines deep learning and gait analysis techniques to analyze wearable sensor data for the purpose of conducting sidewalk surface assessments.
2.  We compared the performance of three different input modalities for the LSTM-based framework and identified the most suitable modality. It was shown that the LSTM networks that took gait feature modalities as the input were able to achieve convergence and robust performance with limited samples compared to the LSTM networks fed with raw data modalities.
3.  We demonstrated that the LSTM networks with gait feature modalities could outperform conventional machine-learning-based methods in the problem domain.
4.  We showed that post-processing on several consecutive per-stride predictions of LSTM networks fed with single-stride gait feature modalities to generate final predictions on a larger segment could improve classification performance.

The rest of the paper is organized in the following manner: We first discuss a review of related works. Next, we present the research methodology utilized in this work, followed by the experiments section where the experimental setup and results are presented. Lastly, a discussion on the findings and implications of this study is provided.

This paper was based on Chapter 4 of the first author's master's thesis [13].

## 2. Related Works

### 2.1. Conventional Methods for Sidewalk Assessments

Some of the prevalent methods employed to maintain sidewalk facilities have relied on conducting pedestrian surveys, interviews and self-reporting [8]. However, these methods are subjective, rely on pedestrians' voluntary participation, lack reliability and are inadequate in providing a comprehensive analysis of sidewalk defects [8]. Governmental agencies also often conduct field inspections carried out by trained inspectors to identify regulatory violations [9]. Studies have also explored new ways to enhance conventional sidewalk assessment practices. Sousa et al. presented a multi-criteria sorting methodology to evaluate sidewalk performance [14]. Meanwhile, Corazza et al. proposed an evaluation index for sidewalk conditions that was derived from the standardized pavement condition index (PCI) used for roads and airports [15]. Nonetheless, these methods require significant labor and financial resources [9]. Additionally, scaling these methods for larger cities poses considerable challenges.

### 2.2. Machine Learning Methods with Hand-Crafted Features for Automated Sidewalk Assessments

With the rapid progress in sensing technologies, multiple research studies have shown the feasibility of detecting various sidewalk features or defects by utilizing wearable sensors to measure pedestrians' physiological responses [8,16,17]. Studies have also examined the impacts of irregular and uneven surfaces on human walking patterns [18–22]. However, these studies were inference-based, aiming to uncover relationships between signal magnitude or specific gait parameters and surface conditions or the built environments.

Consequently, to stimulate advancements in automated sidewalk assessments, few studies have explored the automated assessment of road or surface conditions that integrate human bodily responses utilizing robust machine learning or deep learning techniques. One study [23] utilized features extracted from raw acceleration data gathered from pedestrians' smartphones to train a conventional machine learning model to detect barriers and obstacles in a large area. Takahashi et al. [24] proposed a step-classifying algorithm to detect steps based on features extracted from $x$-axis accelerations gathered from smartphones affixed onto cyclists. Kobayashi et al. predicted different sidewalk surface types using random forest and acceleration data from smartphones stored in pedestrians' front pockets [25]. In

our previous study [10], we proposed a traditional machine learning approach that analyzes hand-crafted accelerometer-based gait features extracted from a right-ankle sensor.

### 2.3. Deep Learning Methods for Automated Sidewalk Assessments

Numerous advanced deep-learning-based approaches have been developed for the automated evaluation of roadways or sidewalks utilizing infrastructure-based data such as images, videos or GIS technologies [26–32]. In one study [33], acceleration data from smartphones mounted on vehicles were used to train deep learning models for road surface monitoring and pothole detection. However, determining the presence of poor sidewalk conditions relies on how individuals interact with sidewalks [34], and such interactions can vary in different contexts [35]. Infrastructure-based data and data collected from moving vehicles, nonetheless, fail to address how the human body reacts to external walking environments [36]. Individuals respond differently to the same surroundings based on their characteristics; therefore, their variability must be analyzed [8].

Limited studies have applied deep neural networks to analyze human responses for walking surface condition detection. Kim et al. verified the effectiveness of a cascaded LSTM-based deep recurrent neural network method to classify abnormal and normal gaits, and demonstrated that the ratio of abnormal gaits could indicate the existence of an environmental barrier to walkability, as they were highly correlated [37]. The confirmed relationship between the ratio of abnormal gaits and the presence of an environmental barrier is in line with our observation of a higher rate of disrupted gaits on irregular walking surface segments. However, the focus of their study on the environmental barrier was on surrounding facilities instead of sidewalk conditions. Hu et al. demonstrated the effectiveness of using deep learning approaches to analyze data from six wearable sensors placed on different body locations to identify irregular walking surfaces [38]. However, the surfaces that were tested in the study did not accurately reflect the usual uneven surfaces found on sidewalks in neighborhoods. Furthermore, deploying sensors at various locations for continuous monitoring in real-time is impractical due to the computational demands of processing data from multiple locations [39], as well as the discomfort caused by wearing multiple sensors. Kobayashi et al. also expanded their study to propose a deep learning approach by training a convolutional neural network (CNN) with acceleration data and window-based features extracted from smartphones [40]. The limitation of this approach is that the models trained with features extracted from smartphone data using a sliding window were unable to be generalized to individual differences.

To address the gaps in existing works and to improve the prediction performance of the traditional machine-learning-based classification framework from our previous study [10], we introduced a novel deep-LSTM-based classification framework to automatically detect irregular walking surfaces with a single wearable sensor in this paper. In our previous study [10], we identified the right ankle as the ideal location for sensor placement for detecting irregular walking surfaces. Hence, in this paper, we used the acceleration data collected from a right-ankle sensor from our previous experiment and the traditional machine learning approach as the benchmark to validate the feasibility of our proposed deep LSTM framework.

## 3. Materials and Methods

### 3.1. Proposed Framework

Figure 1 illustrates the overview of the proposed framework. The data collection and labelling process is described in Section 3.2. The measured raw acceleration data from the previous study were pre-processed in three different ways to generate three different modalities to feed distinct LSTM networks, because the workflow for implementing a recurrent deep learning method differs based on the input modality, which is discussed in Section 3.3. Next, hyper-parameters for each LSTM network had to be tuned to determine the optimal network structure for each input modality. The setup is discussed in Section 4.1. Finally, all LSTM networks were then evaluated with the leave-one-subject-out test set

procedure. All five scenarios of irregular walking surfaces were consolidated into a single class labeled as "Irregular", framing the problem as a binary classification task.

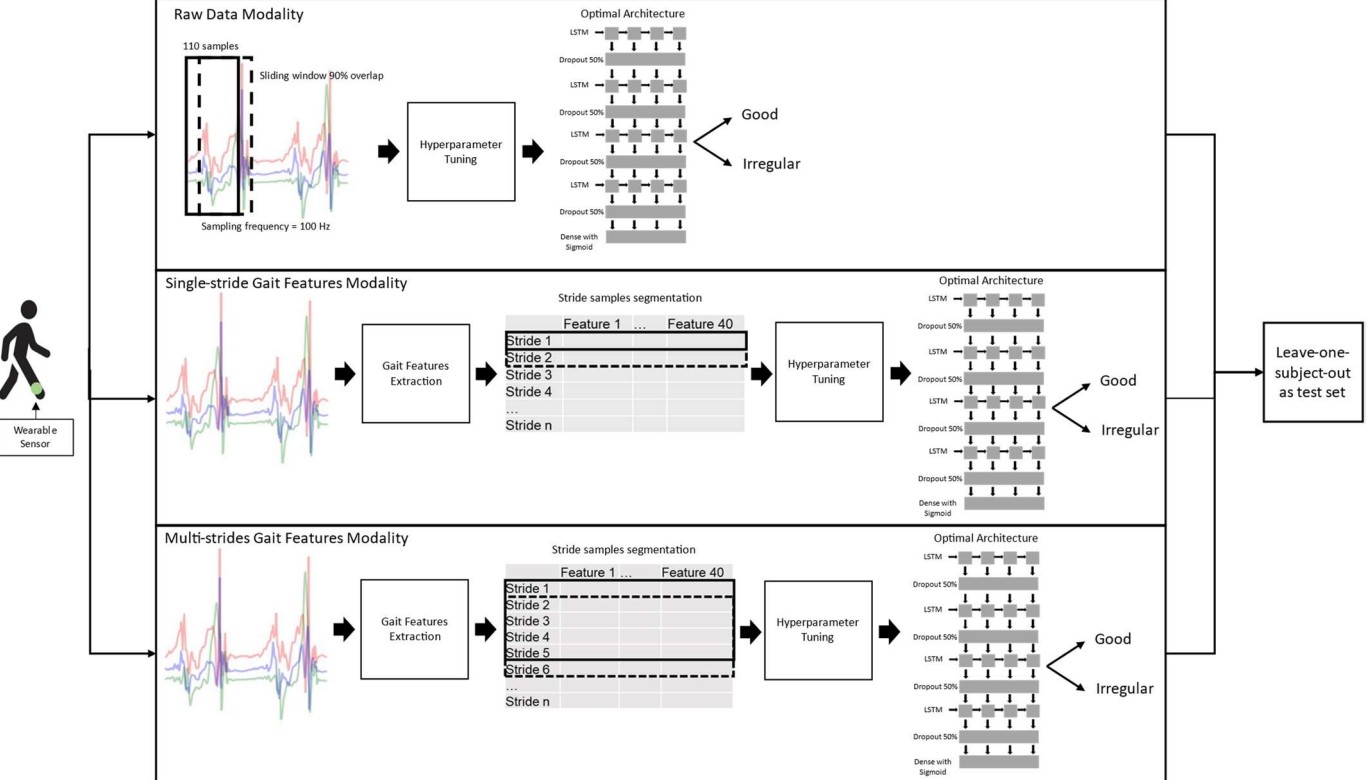

**Figure 1.** Overview of the proposed framework.

### 3.2. Data Collection

We used the data collected from the accelerometer positioned at the right ankle of 12 subjects from our previous experiment [10] for this study, since the right ankle was determined to be the most suitable location to identify irregular walking surfaces. To collect the acceleration data, we conducted the experiment at The Peter Kiewit Institute of the University of Nebraska at Omaha. We selected a segment of a well-paved, smooth and leveled walking route in that area and denoted the starting and ending points. Then, four irregular walking surface segments were set up within that path: a segment covered in grass, an object-obstructed segment, an uneven segment and a segment covered with debris. These segments represented irregular sidewalk walking surfaces that are highly probable in less walkable neighborhoods in the real-world. The good and irregular walking segments are shown in Figure 2.

We recruited twelve healthy participants, comprising eight males and four females, for this study. A tri-axial accelerometer was affixed at the right ankle on each subject, as shown in Figure 3. It was affixed to the outer side of the shoe of each participant. We measured the linear acceleration along the X, Y and Z axes of the subject's body with Mbient sensors (MetaMotionR, Mbient Lab, San Francisco, CA, USA) that were configured to capture data at a rate of 100 Hz. The subjects were directed to walk at their usual pace along the path, starting from the initial point, proceeding towards the endpoint and then returning to the starting point along the reverse path. Video recordings were taken throughout the experiment for each subject to help facilitate data labeling.

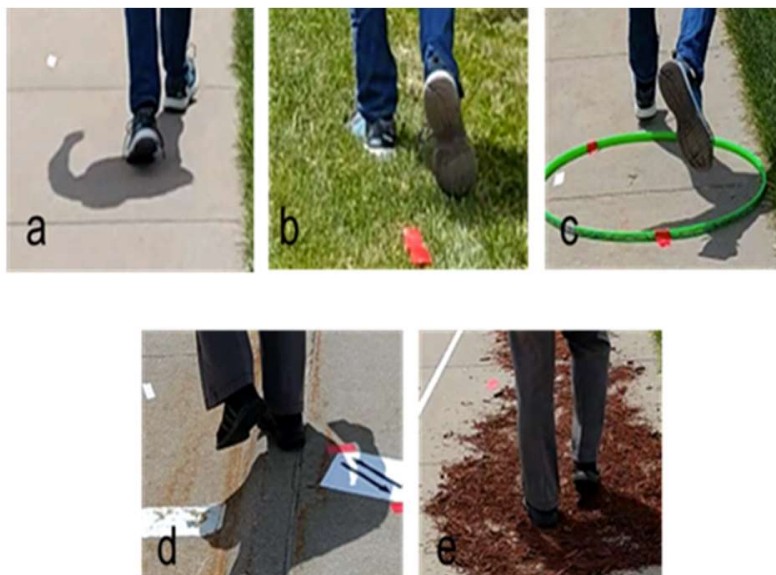

**Figure 2.** Good and irregular walking surfaces; (**a**) well-paved (good), even; (**b**) grassed surface (irregular); (**c**) surface obstructed with objects (irregular); (**d**) uneven (irregular); and (**e**) surface covered with debris (irregular).

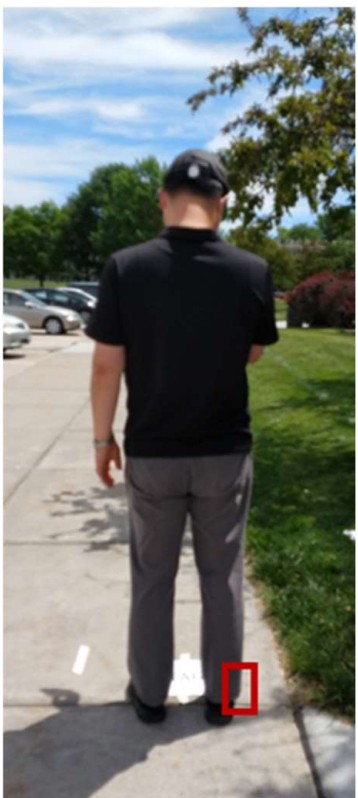

**Figure 3.** Placement of sensor at the right ankle of a subject.

Figure 4 displays the raw acceleration measurements captured with the right-ankle sensor of a subject. The figure visualizes an individual's gait for five seconds, which is equivalent to 500 data points. The raw acceleration data points were categorized by cross-referencing with the experiment videos.

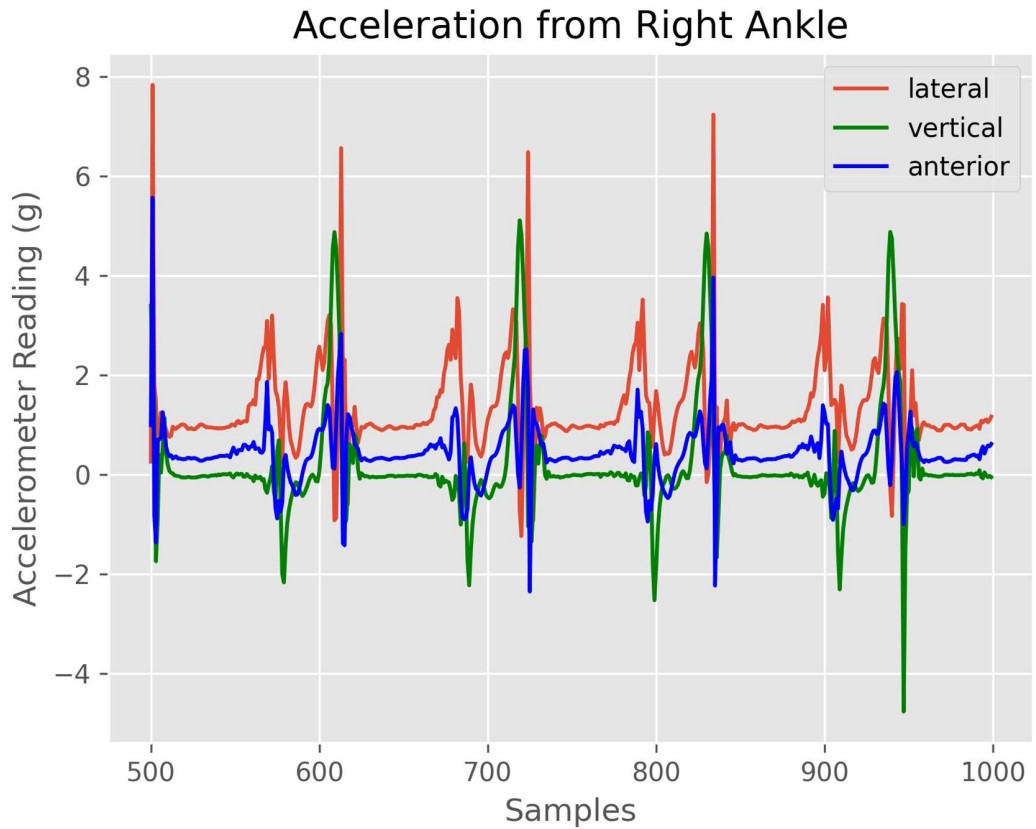

**Figure 4.** Five seconds of raw walking acceleration signal of a subject captured with a sensor positioned at the right ankle.

### 3.3. Data Pre-Processing

The acceleration data from the accelerometer positioned at the right ankle of subjects were analyzed to discriminate between good and irregular walking surfaces. As shown in Figure 5, the X, Y and Z axes of the right ankle accelerometer captured acceleration data corresponding to motion in the vertical (V), antero-posterior (AP) and medio-lateral (ML) directions, respectively.

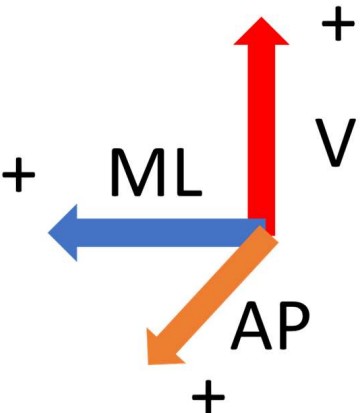

**Figure 5.** Accelerometers' directions for right ankle.

To compare how the three different modalities affected the classification performance of the proposed network, three distinct LSTM models had to be trained and fed with different data dimensions. The raw acceleration data needed to be pre-processed as described in the following section.

### 3.3.1. Raw Data Modality

We collected a total of 646,500 samples of right-ankle raw acceleration data, with 426,500 samples falling on good walking surfaces and 220,000 samples on irregular walking surfaces, as shown in Table 1. Since taking large values directly to train a LSTM network could lead to large gradient updates that would result in a slower convergence or even divergence of the training process, we normalized our input data to have a standard deviation of 1 and a mean of 0. The input to be fed into the network had to be fixed-size sequential data, which was a short time series segmented from the raw acceleration signal. Therefore, the acceleration data were sampled into each segment with a 1.10 s fixed-width sliding window, which is equivalent to 110 data points, with a 90% overlap between them. Since the acceleration data had X, Y and Z dimensions, the sliding window would sample data points from all three dimensions into each segment, resulting in a three-dimensional input segment vector.

**Table 1.** Distribution of class labels of raw acceleration data.

| Label | Count | Percentage |
|---|---|---|
| Good | 426,500 | 66% |
| Irregular | 220,000 | 34% |
| Total | 646,500 | 100% |

This window size was chosen because each window should have included at least one complete period [41], which was one stride. The average stride time for all subjects was 1.10 s. The 90% overlap rate was chosen because it would result in more segmented data samples due to the limitation of our sample size. Additionally, an overlap between windows would carry information from the previous window.

### 3.3.2. Single-Stride and Multi-Stride Gait Feature Modalities

For the training of the proposed network using gait features, we adopted the same gait features employed in our previous study [10]. The extracted features were shown to be useful and robust in training predictive models in previous mobility studies [42,43], since these features captured the biomechanical gait characteristics of various phases of a person's gait cycle. To extract the gait features from raw acceleration data, the first step involved segmenting the raw acceleration data into strides using the AP directional acceleration of the accelerometer [42]. Next, we computed 20 base gait features from the segmented strides before computing the variability of those gait features between strides, thereby resulting in a total of 40 gait features [10]. A complete description of the base gait features can be found in Table 2, while the distribution of stride samples for each label is shown in Table 3.

**Table 2.** Description of extracted base features.

| Gait Feature | Gait Feature Description | Gait Feature Formula |
|---|---|---|
| VM | Magnitude of the vector for the entire stride | $\sqrt{ML^2 + V^2 + AP^2}$ computed from the acceleration of the three axes for the entire stride |
| VM5 | Magnitude of the vector for the first 5% portion of the stride | $\sqrt{ML^2 + V^2 + AP^2}$ computed from the acceleration of the three axes for the first 5% portion of the stride |
| LVM | Magnitude of the vector of the ML axis for the entire stride | $\sqrt{ML^2}$ computed from the acceleration of the ML axis for the entire stride |
| VVM | Magnitude of the vector of the V axis for the entire stride | $\sqrt{V^2}$ computed from the acceleration of the V axis for the entire stride |

**Table 2.** *Cont.*

| Gait Feature | Gait Feature Description | Gait Feature Formula |
|---|---|---|
| AVM | Magnitude of the vector of the AP axis for the entire stride | $\sqrt{AP^2}$ computed from the acceleration of the AP axis for the entire stride |
| VMD | Magnitude of the vector for the double stance | $\sqrt{ML^2 + V^2 + AP^2}$ computed from the acceleration of the three axes for the $\pm 10\%$ portion surrounding the heel strike event |
| LVMD | Magnitude of the vector of the ML axis at the time of a double stance | $\sqrt{ML^2}$ computed from the acceleration of the ML axis for the $\pm 10\%$ portion surrounding the heel strike event |
| VVMD | Magnitude of the vector of the V axis at the time of a double stance | $\sqrt{V^2}$ computed from the acceleration of the V axis for the $\pm 10\%$ portion surrounding the heel strike event |
| AVMD | Magnitude of the vector of the AP axis at the time of a double stance | $\sqrt{AP^2}$ computed from the acceleration of the AP axis for $\pm 10\%$ portion surrounding the heel strike event |
| VM30 | Magnitude of the vector for the mid-stance | $\sqrt{ML^2 + V^2 + AP^2}$ computed from the acceleration of the three axes for the 30% portion of the gait cycle |
| LVM30 | Magnitude of the vector of the ML axis at the time of the mid-stance | $\sqrt{ML^2}$ computed from the acceleration of the ML axis for the 30% portion of the gait cycle |
| VVM30 | Magnitude of the vector of the V axis at the time of a mid-stance | $\sqrt{V^2}$ computed from the acceleration of the V axis for the 30% portion of the gait cycle |
| AVM30 | Magnitude of the vector of the AP axis at the time of a mid-stance | $\sqrt{AP^2}$ computed from the acceleration of the AP axis for the 30% portion of the gait cycle |
| LHM | Magnitude of heel strike of the ML axis | Max (ML) at the heel strike |
| LHSD | Standard deviation of the acceleration of the ML axis for the first 10% portion of the stride | Std (ML) computed from the acceleration of the ML axis for the first 10% portion of the stride |
| VHM | Magnitude of heel strike of the V axis | Max (V) at the heel strike event |
| VHSD | Standard deviation of the acceleration of the V axis for the first 10% portion of the stride | Std (V) computed from the acceleration of the V axis for the first 10% portion of the stride |
| AHM | Magnitude of heel strike of the AP axis | Max (AP) at the heel strike event |
| AHSD | Standard deviation of the acceleration of the AP axis for the first 10% portion of the stride | Std (AP) computed from the acceleration of the AP axis for the first 10% portion of the stride |
| ST | Stride time | Duration of two consecutive heel strike events |

**Table 3.** Distribution of class labels of stride samples.

| Label | Count | Percentage |
|---|---|---|
| Good | 3774 | 65% |
| Irregular | 1995 | 35% |
| Total | 5769 | 100% |

The stride samples were also normalized to have a standard deviation of 1 and a mean of 0. Subsequently, the stride samples were also segmented into a fixed-size data sequence. To prepare the stride samples for the training of the single-stride gait feature modality, since one sample was equivalent to one stride, a sliding window with the width of one sample with no overlap was used to segment the data, as shown in Figure 6. Since there were 40 features, the input segment vector would contain 40 dimensions.

In our previous study [10], we observed that gaits were not always disrupted when subjects traversed irregular walking surfaces. Abnormal gaits would intermittently occur alongside normal gaits. Therefore, compared to walking on a good walking surface, there was a higher rate of interrupted gaits. Based on this observation, we hypothesized that aggregating multiple consecutive strides to train an LSTM network would improve

classification performance. This led to the experimentation of the multi-stride gait feature modality. For this modality, a larger sliding window width of five samples was used to segment the data, which was equivalent to five strides, with an 80% overlap, as illustrated in Figure 7. The input segment vector had the shape of 40 dimensions as well.

**Figure 6.** Segmentation of gait feature data into fixed-size segments of one sample with a sliding window.

**Figure 7.** Segmentation of gait feature data into fixed-size segments of five samples with a sliding window with 80% overlap.

*3.4. Proposed Architecture*

Since the architecture of a LSTM model is input-dependent, the number of layers and the optimal network structure for each of the three modalities differed. For ease of reference, the LSTM networks trained with raw data, single-stride gait features and multi-stride gait features were referred to as LSTM-Raw, LSTM-Features-1 and LSTM-Features-5, respectively. The tuning methodology is delineated in Section 4.1.1.

The resulting optimal architectures turned out to be identical for all three networks, as illustrated in Figure 8. It consisted of four layers of a LSTM layer with dropout layers in between each LSTM layer. At the end of the architecture was a dense layer, followed by a Sigmoid activation function that yielded the probability of the pre-processed sample being an irregular class.

However, the optimal number of neurons per LSTM layer for LSTM-Raw was 224, while the optimal number of neurons for both LSTM-Features networks was 100. This meant that the dimensionality of the input vector per time-step fed into a LSTM layer in LSTM-Raw was three and the output vector, which was also the hidden state vector, of the layer had a dimensionality of 224. On the other hand, the dimensionality of the input vector per time-step fed into a LSTM layer in the LSTM-Features networks was 40 and would output a hidden state vector with a dimensionality of 100.

Stacking multiple LSTM layers could transform the raw acceleration and gait feature inputs into a more abstract representation that enabled the network to learn the complex relationships between the inputs and walking surface conditions [44]. We kept the number

of neurons the same for each layer when tuning, because using the same number of neurons for all layers performed generally better than increasing or decreasing the neurons down the network [45]. As shown in Figure 8, the pre-processed data were fed into the stacked LSTM layers to extract temporal features. The temporal features were then processed with the dense layer, followed by a sigmoid function to obtain the probability of the pre-processed sample being an irregular class.

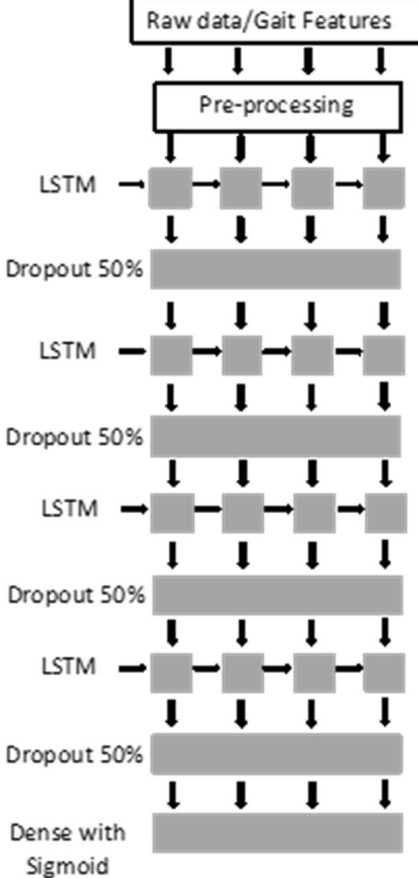

**Figure 8.** Stacked LSTM architecture.

3.4.1. LSTM Layer

LSTM is a specialized variant of the recurrent neural network (RNN), and it was introduced by Hochreiter et al. [46]. Unlike a traditional neural network, which is incapable of capturing temporal information as the nodes in each layer are disconnected, the RNN compensates for that limitation by mapping input sequences to output sequences. However, the RNN has the disadvantage of recognizing data for only short periods of time due to the vanishing/exploding gradient [47]. The LSTM solves the problem of the RNN with special memory cells that can solve complex, artificial, long-time-lag tasks, which resulted in it being an appropriate solution for our problem, since acceleration data are time series data [46]. The LSTM network introduced a new mechanism called a memory cell.

A memory cell is regulated with three main gates: an output gate, an input gate and a forget gate. The output gate protects other neurons from being perturbed by a currently irrelevant memory cell state, while the input gate determines what input can alter the memory cell state [46]. Lastly, the forget gate is responsible for gathering prior data and what past information to forget from the cell's previous state [46].

A LSTM memory cell would take in an input $x_t$ at time $t$, the hidden state $h_{t-1}$ and the cell state $c_{t-1}$ from the previous LSTM memory cell, and would output the current hidden state $h_t$ plus the memory cell state $c_t$. $U$, $W$ and $V$ denote the weight matrices,

while $b$ represents the bias vector. The forward pass of the LSTM unit could be expressed as follows:

$$f_t = \sigma(W_f x_t + U_f h_{t-1} + b_f) \tag{1}$$

$$i_t = \sigma(W_i x_t + U_i h_{t-1} + b_i) \tag{2}$$

$$\tilde{c}_t = tanh(W_c x_t + U_c h_{t-1} + b_c) \tag{3}$$

$$c_t = i_t \circ \tilde{c}_t + f_t \circ c_{t-1} \tag{4}$$

$$o_t = \sigma(W_o x_t + U_o h_{t-1} + V_o c_t + b_o) \tag{5}$$

$$h_t = o_t \circ tanh(c_t) \tag{6}$$

where $\circ$ represents the element-wise multiplication, while $f_t$, $i_t$, $\tilde{c}_t$ and $o_t$ represent the forget, input, cell, output gates' outputs, respectively. $c_t$ and $h_t$ are passed on to the next time step.

As mentioned above, the LSTM-Raw network had 224 memory cells for each LSTM layer and would take in a three-dimensional input vector $x_t$ per time-step. Each of the LSTM layers would produce a hidden state output vector $h_t$ with 224 dimensions. As for the LSTM-Features networks, it would take in an input vector $x_t$ with 40 dimensions per time step. As each LSTM layer had 100 memory cells, each layer would generate a hidden state output vector $h_t$ with 100 dimensions.

### 3.4.2. Dropout Layer

Dropout is a technique commonly used to prevent overfitting in a deep neural network with a vast number of parameters [44]. Limited training data in our study would lead to overfitting, since we were using multiple LSTM layers. The main idea is to randomly drop input units into the next layer during training, which prevents the units from co-adapting too much [44]. We set the dropout rate to be 50%. This meant that when the hidden state vector output by the previous LSTM layer was fed to the dropout layer, the dropout layer would randomly set 50% of the hidden state vector dimensions to 0, which would be the input vector into the next layer.

### 3.4.3. Dense Layer

The dense layer and a Sigmoid function were necessary to process feature representations extracted with the LSTM layers to complete our classification task. The dense layer was also a fully connected layer, which meant that hidden state vectors generated by each LSTM neuron after being randomly dropped out were being fed to every node in the dense layer.

Since the output of the dense layer was not interpretable, it was fed into a Sigmoid function so that it could be converted into a probability of the given sample being an irregular class. The formula for Sigmoid is as follows:

$$S(x) = \frac{1}{1 + e^x} \tag{7}$$

where $x$ is the one-dimensional output vector of the densely connected layer.

## 4. Experiments Section

### 4.1. Experimental Setup

#### 4.1.1. Hyper-Parameter Tuning

To determine the optimal network structure for each of the three modalities, the number of neurons and layer hyper-parameters had to be tuned independently for each modality. We first split the samples for each modality into training and test data subsets

using an 80:20 split. The training data subset was further split into 80% for training and 20% for validation during the model fitting, so that the model learning behavior could be monitored through training and validation learning curves to avoid underfitting or overfitting. The test set was used to evaluate the structure.

We tuned the number of layer and number of neuron hyper-parameters with the automated tuning package in Keras Tuner [48]. The hyper-parameters were optimized using random searches to minimize validation loss in 30 maximum trials.

Neural networks are commonly trained using mini-batch processing. According to Keskar et al. [49], using a larger batch size would lead to a less generalizable model. Hence, upon determining the network structure, we also examined the effect of the batch size on the classification performance to determine the most effective batch size.

### 4.1.2. Model Implementation

The proposed LSTM structure was implemented with Keras libraries [48], written in Python and utilizing Tensorflow [50] as the backend. All the LSTM models with different modalities were trained in a supervised manner. Each layer's weights and biases were initialized with randomly selected values. The LSTM parameters were optimized through minimizing the binary cross-entropy loss function. The binary cross-entropy loss function measures the distance between the predicted probability and the actual class. The binary cross-entropy loss for a batch of N samples was defined as:

$$L = -\sum_{i=1}^{N} y_i log(\hat{y}_i) + (1 - y_i)log(1 - \hat{y}_i) \tag{8}$$

where $y_i$ denotes the truth value, taking the value 1 or 0, and $\hat{y}_i$ is the predicted Sigmoid probability.

An optimizer used to update the network parameters to minimize the loss function needed to be selected as well. We chose the Adam optimizer, since it was shown that it had the best fitting effect for the purpose of human activity recognition [41]. A learning rate of 0.0001 was used to control the extent of the parameter updates.

### 4.1.3. Evaluation Metrics

We had a disproportionate sample of irregular classes in our dataset, as shown in Tables 1 and 3. Hence, we used the area under the receiver operating characteristic (ROC) curve (AUC), which described the relationship between the false positive rate and true positive rate, as our performance metric because it is unaffected by skewed data [51,52].

Furthermore, once we determined the optimal network structure for all three modalities, to verify the generalizability of the models to subject differences, we employed the leave-one-subject-out test set procedure for the evaluation of each modality. During each iteration of the protocol, to avoid overfitting for each model, the best epoch to halt training was determined by utilizing the early-stopping strategy [45]. We initially set the training epoch to an arbitrary large number of 200 in Keras and configured the early-stopping criterion to halt training when there was no improvement in the validation AUC after 15 epochs. The optimal number of epochs for each respective model was then used to retrain the models, and the models were evaluated with the test set.

### 4.1.4. Traditional Machine Learning Baseline Method

We verified the effectiveness of the proposed approach by benchmarking the approach against the shallow machine learning method that was presented in our previous study [10]. We used an SVM model as our baseline shallow machine learning model for this experiment. The workflow of training an SVM model is shown in Figure 9.

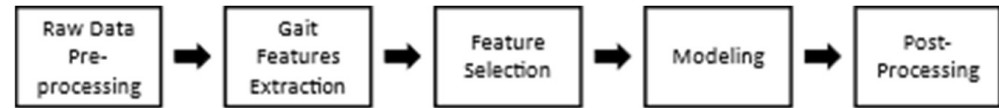

**Figure 9.** Workflow of training an SVM model with gait features.

We pre-processed the raw data and performed stride segmentation prior to extracting the 40-dimensional gait features described in Table 2, which was identical to the pre-processing steps performed for the gait feature modalities delineated in Section 3.3.2. The optimal feature subset was, subsequently, identified from the extracted gait feature set to train the SVM model. The trained SVM model would generate predictions on a stride-by-stride basis to discern whether a stride occurred on an irregular surface or not.

Our preliminary study also demonstrated that the classification performance of the SVM model could be improved with post-processing on the per-stride predictions of the SVM, since there were more frequent gait interruptions on irregular walking surfaces compared to the good walking surfaces. We segmented the per-stride predictions by sampling them into five consecutive strides for each segment using a sliding window with the step size of one stride and averaging the predicted probability to obtain the final prediction of that segment. The ground truth of the segment was based on the majority of stride labels in the segment. Furthermore, for a complete comparison, we also conducted the same pre-processing steps on LSTM-Features-1's per-stride predictions and compared the classification performance of the per-stride and post-processing predictions between our proposed approach and the shallow SVM approach.

*4.2. Experimental Results*

4.2.1. Selected Architectures

After hyper-parameter tuning, the resulting optimal structure for LSTM-Raw was four hidden layers with 224 neurons at each layer. For both the LSTM-Features-1 and LSTM-Features-5 modalities, the optimal architecture was identical, consisting of a configuration with four hidden layers and 100 neurons at each layer. Tables 4–6 summarize the list of the selected hyper-parameters for all three models, as well as the total number of trainable parameters for each network.

**Table 4.** List of selected hyper-parameters for LSTM-Raw.

| Stage | Hyper-Parameters | Selected Values |
|---|---|---|
| Data Pre-Processing | Window Size | 110 |
| | Step Size | 11 |
| Architecture | LSTM_1 Neurons | 224 |
| | LSTM_2 Neurons | 224 |
| | LSTM_3 Neurons | 224 |
| | LSTM_4 Neurons | 224 |
| | Dropout Rate | 0.5 |
| Training | Optimizer | Adam |
| | Batch Size | 32 |
| | Learning Rate | 0.0001 |
| Total Trainable Parameters | | 1,411,425 |

**Table 5.** List of selected hyper-parameters for LSTM-Features-1.

| Stage | Hyper-Parameters | Selected Values |
|---|---|---|
| Data Pre-Processing | Window Size | 1 |
| | Step Size | 1 |

**Table 5.** *Cont.*

| Stage | Hyper-Parameters | Selected Values |
|---|---|---|
| Architecture | LSTM_1 Neurons | 100 |
| | LSTM_2 Neurons | 100 |
| | LSTM_3 Neurons | 100 |
| | LSTM_4 Neurons | 100 |
| | Dropout Rate | 0.5 |
| Training | Optimizer | Adam |
| | Batch Size | 32 |
| | Learning Rate | 0.0001 |
| Total Trainable Parameters | | 297,701 |

**Table 6.** List of selected hyper-parameters for LSTM-Features-5.

| Stage | Hyper-Parameters | Selected Values |
|---|---|---|
| Data Pre-Processing | Window Size | 5 |
| | Step Size | 1 |
| Architecture | LSTM_1 Neurons | 100 |
| | LSTM_2 Neurons | 100 |
| | LSTM_3 Neurons | 100 |
| | LSTM_4 Neurons | 100 |
| | Dropout Rate | 0.5 |
| Training | Optimizer | Adam |
| | Batch Size | 32 |
| | Learning Rate | 0.0001 |
| Total Trainable Parameters | | 297,701 |

### 4.2.2. Effect of Batch Size

We tuned our hyper-parameters using the default batch size of 32, which is widely considered as a practical and effective value [45]. Here, we examined the effect of batch sizes of 16, 32, 64, 128 and 256 on the classification performance for all three architectures. Table 7 shows the classification results of different batch sizes for LSTM-Raw, LSTM-Features-1 and LSTM-Features-5. It was noticeable that boosting the batch size resulted in a reduction in the prediction accuracy of each LSTM, which was in line with the conclusions drawn by Keskar et al. [49]. At a batch size of 16, the classification results were similar to those attained at the batch size of 32. Since a larger batch size yielded faster computation [45], we picked 32 as the best batch size and used it for the training of our LSTM models.

**Table 7.** Classification results of different batch sizes across all LSTM models.

| Batch Size | LSTM-Raw (AUC) | LSTM-Features-1 (AUC) | LSTM-Features-5 (AUC) |
|---|---|---|---|
| 16 | 0.92 | 0.86 | 0.93 |
| 32 | 0.92 | 0.86 | 0.93 |
| 64 | 0.91 | 0.85 | 0.92 |
| 128 | 0.91 | 0.85 | 0.91 |
| 256 | 0.90 | 0.85 | 0.90 |

### 4.2.3. Leave-One-Subject-Out Assessment

The classification outcomes for every test subject of the leave-one-subject-out assessment for each LSTM model and the SVM baseline model are summarized in Table 8.

**Table 8.** Classification results for each test subject for all LSTM models and SVM.

| Test Subject | LSTM-Raw (AUC) | LSTM-Features-1 (AUC) | LSTM-Features-5 (AUC) | SVM (AUC) |
|---|---|---|---|---|
| A | 0.62 | 0.82 | 0.86 | 0.74 |
| B | 0.60 | 0.84 | 0.80 | 0.81 |
| C | 0.78 | 0.87 | 0.87 | 0.83 |
| D | 0.77 | 0.78 | 0.79 | 0.70 |
| E | 0.85 | 0.84 | 0.85 | 0.85 |
| F | 0.55 | 0.82 | 0.83 | 0.70 |
| G | 0.85 | 0.91 | 0.94 | 0.84 |
| H | 0.70 | 0.77 | 0.84 | 0.87 |
| I | 0.70 | 0.79 | 0.82 | 0.82 |
| J | 0.65 | 0.76 | 0.76 | 0.72 |
| K | 0.51 | 0.71 | 0.72 | 0.75 |
| L | 0.70 | 0.85 | 0.87 | 0.92 |
| Average | 0.69 | 0.81 | 0.83 | 0.80 |

LSTM-Raw was the least efficient, as it exhibited an inferior classification performance compared to the SVM baseline, and had the lowest average AUC. Both the LSTM-Features models slightly outperformed the SVM model, with LSTM-Features-5 achieving the highest average AUC of 83%. These results validated the feasibility of utilizing hand-crafted features to guide the learning of a LSTM network into convergence with the limited sample size and subject count. When comparing the individual subject outcomes between LSTM-Features-1 and LSTM-Features-5, we observed that introducing multiple strides during training in LSTM-Features-5 resulted in an AUC improvement ranging from 1% to 7%, leading to a 2% increase in the average AUC performance. This finding confirmed the hypothesis that incorporating multiple strides during training could enhance the performance of the LSTM-Features model. The LSTM-Features-5 model demonstrated the highest level of robustness, as it was able to consistently attain an AUC of over 71% regardless of the selection of subject data excluded from training. Compared to the LSTM model trained with raw acceleration data, the impact of individual differences on the performance of both the LSTM and SVM models was less noticeable when trained with gait features. For completeness, Figure 10 shows the ROC curve of LSTM-Features-5 with one of the test subject iterations that was closest to the average AUC. The area under the ROC curve was 83%.

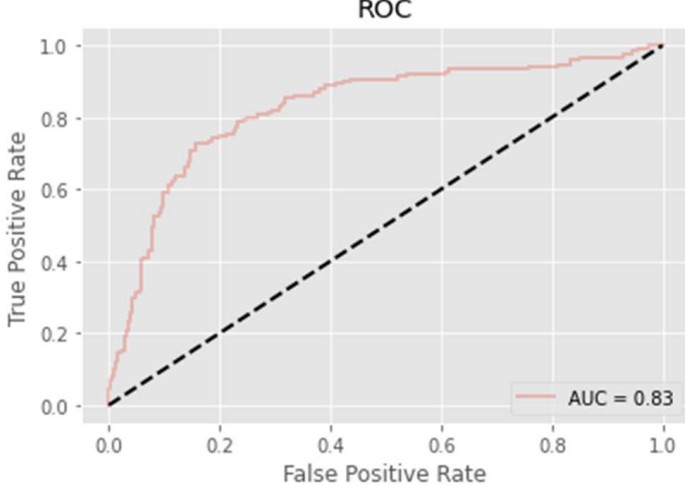

**Figure 10.** The ROC curve of one of the test subject iterations for LSTM-Features-5.

Figures 11 and 12 depict the training process of one of the leave-one-subject-out iterations for LSTM-Features-5. During the network training, the training loss decreased

gradually, while the validation AUC stabilized, indicating that the model achieved convergence. With the early-stopping strategy, the training stopped at epoch 78 when the validation AUC stopped improving after 15 epochs to prevent overfitting. As shown in Figure 11, the gaps between the training and validation curves in the binary cross-entropy loss were small. Figure 12 shows that the gaps between the training and validation curves in terms of the AUC were exceedingly small. These results indicated that the dropout technique was effective in preventing overfitting. After the learning curves stabilized and the model achieved a good fit, training beyond that point could lead to overfitting. To prevent this, the early-stopping strategy was employed.

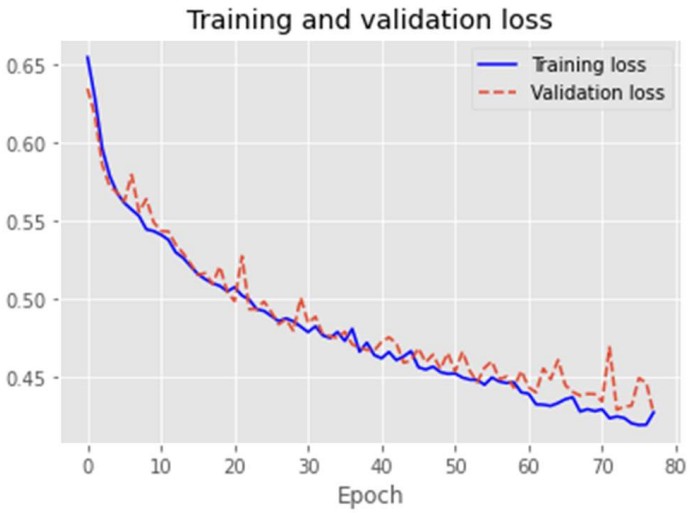

**Figure 11.** Training progress of one of the leave-one-subject-out iterations.

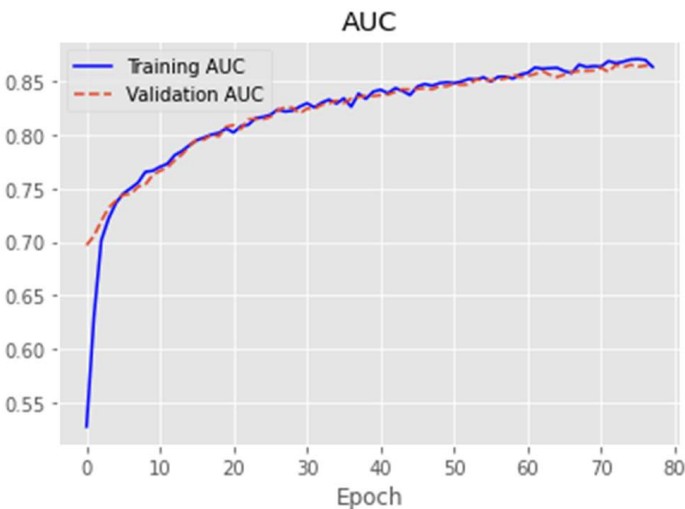

**Figure 12.** Training and validation AUC at each epoch for one of the leave-one-subject-out iterations.

4.2.4. Post-Training Stride Aggregation Post-Processing

The five-stride post-processing classification results for the SVM and LSTM-Features-1 for each test subject are shown in Table 9. The subject-wise classification results of the SVM and LSTM-Features-1 improved across the board with post-processing compared to the per-stride predictions in Table 8. The SVM's average classification performance was approximately 5% higher after post-processing, while LSTM-Features-1's average AUC improved by 7%. Post-processing on the SVM's per-stride predictions resulted in an AUC improvement for all subjects in the range of 1% to 10%, while post-processing on LSTM-Features-1's per-stride predictions led to an AUC improvement in the range of 3% to 9%. Comparing the post-processing results of the SVM and LSTM-Features-1, we

could see that LSTM-Features-1 outperformed the SVM by an average of 2%. This meant that LSTM-Features-1 with post-processing was the most effective model, as it exceeded LSTM-Features-5's classification performance by 5%. This finding suggested that post-training stride aggregations could lead to better classification results, and they could be more effective than incorporating multiple strides during the training stage.

**Table 9.** Five-stride post-processing classification results for each test subject for SVM and LSTM-Features-1.

| Test Subject | SVM (AUC) | LSTM-Features-1 (AUC) |
| --- | --- | --- |
| A | 0.77 | 0.91 |
| B | 0.91 | 0.90 |
| C | 0.88 | 0.92 |
| D | 0.79 | 0.83 |
| E | 0.91 | 0.87 |
| F | 0.73 | 0.89 |
| G | 0.94 | 0.96 |
| H | 0.94 | 0.86 |
| I | 0.89 | 0.87 |
| J | 0.77 | 0.79 |
| K | 0.76 | 0.80 |
| L | 0.95 | 0.91 |
| Average | 0.85 | 0.88 |

## 5. Discussion

In this study, we endeavored to improve upon the traditional baseline support vector machine (SVM) machine learning method from our previous study [10] by leveraging the LSTM network's automated feature learning and knowledge distillation abilities. We first examined how three input modalities impacted the performance of the LSTM network and verified their effectiveness by using the baseline model as the benchmark. The results showed that the LSTM trained with single-stride and multi-stride gait features improved the overall performance compared to the baseline model, as the model's ability was enhanced to capture relevant patterns. This also confirmed the feasibility of the proposed approach.

The leave-one-subject-out test set assessment evaluated the robustness of the models to individual differences. Our results, displayed in Section 4, showed that the LSTM models trained with gait features outperformed the LSTM model with raw data and were more generalizable to individual walking pattern differences. As presented in Table 8, the prediction results of LSTM-Raw were considerably low, with only 51% AUC for subject K and 55% AUC for subject F, which were deemed unacceptable. The LSTM-Feature models, on the other hand, could produce satisfactory outcomes for both subjects. The analysis presented here examined the causes of the inadequate performance of LSTM-Raw for Subjects F and K. Figures 13 and 14 illustrate the boxplots of two top gait features identified in our previous study, VM and VMD, and the distribution of values of each subject for those gait features. Subjects F and K were distributed towards the lower fence of the boxplots.

Figure 15 illustrates the walking patterns of all subjects by combining those two features on a scatterplot to examine if clusters were formed. Three distinct clusters could be observed in the plot. Subjects F and K's walking patterns were distinctively different from the general pattern of the other two clusters. The plausible explanation for the less than satisfactory performance of the LSTM network trained with raw data was that deep learning models are negatively affected by noise effects in a raw signal, limited sample size and subject variety [53,54]. The limited subject sample size and variety of walking patterns led to the less comprehensive learning of feature representation from raw signals. This constraint was mitigated by using hand-crafted gait features to train the LSTM network so that the LSTM model could focus on identifying discriminative features [53]. Furthermore, raw signals measured with the accelerometer were noisy, therefore, producing large intra-class variances. Acceleration data that varied significantly for the same walking surface

condition led to overfitting. Utilizing gait features would reduce noise and overfitting, hence, making the model more generalizable. Additionally, the LSTM networks trained with gait features also required less training time and computational cost, as the optimal models had fewer parameters to train. This is highly desirable for the development of a real-time irregular walking surface detection application.

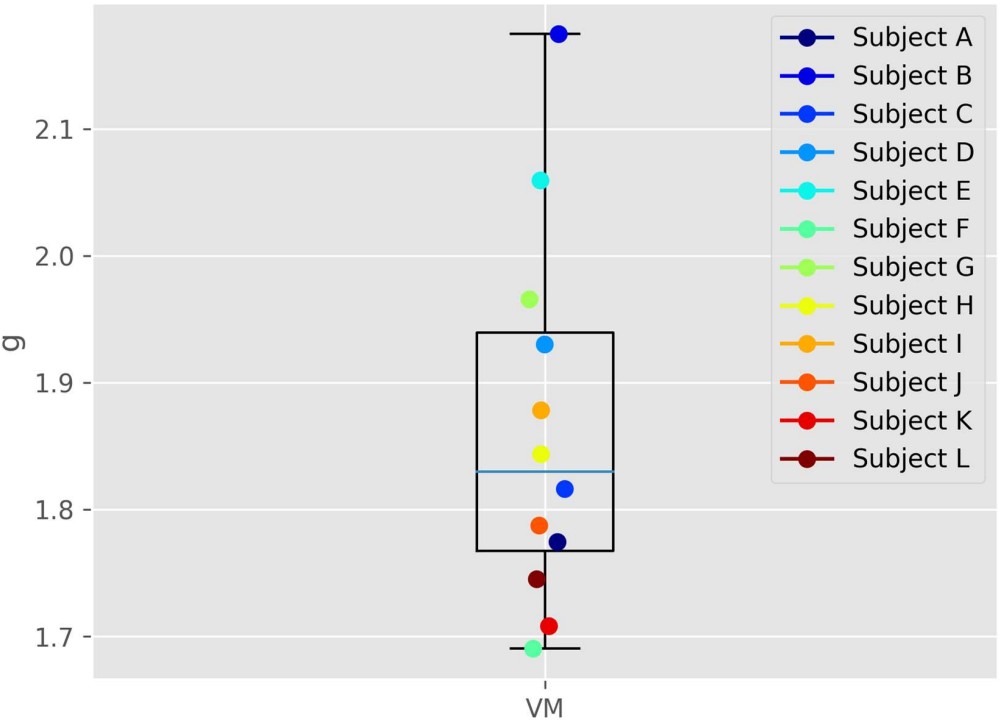

**Figure 13.** Distribution of mean vector magnitude (VM) of each subject on boxplot.

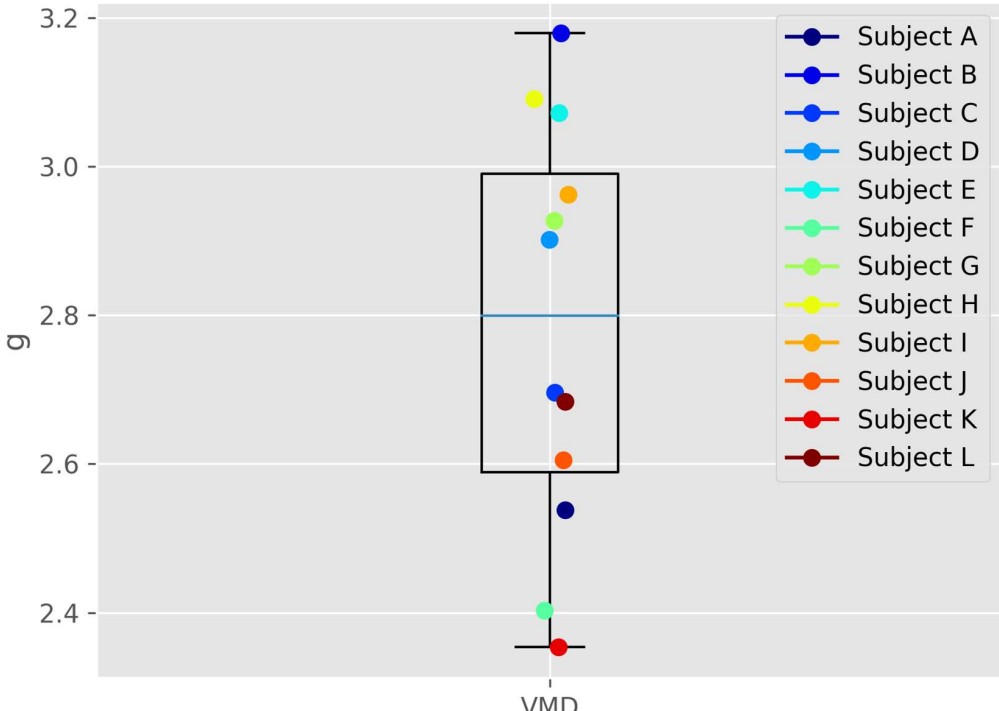

**Figure 14.** Distribution of mean vector magnitude for double-stance phase (VMD) of each subject on boxplot.

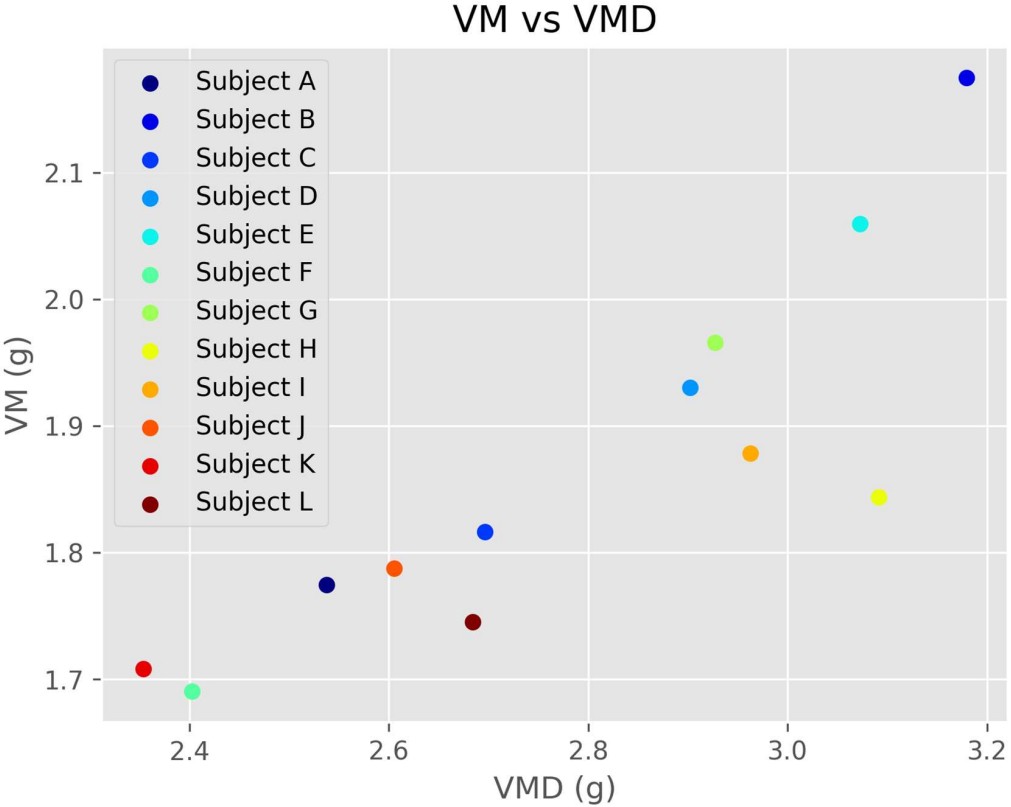

**Figure 15.** Subject distribution based on VM and VMD features in scatterplot.

The experimental results also indicated that post-processing on several consecutive per-stride predictions and incorporating multiple strides per window in the training stage to generate a final prediction for a larger segment could improve the classification performance. This observation was consistent with Kobayashi et al.'s research, which showed that accuracy improved when segmenting smartphone acceleration into a larger window size for convolutional neural network training for a sidewalk type estimation [40]. If the recommended LSTM technique was to be applied in a real-world scenario, it would be essential to capture at least five seconds of data, since a typical stride time for pedestrians is approximately 1.10 s.

Conventional practices for sidewalk condition assessments, which depend on trained experts from governmental agencies or the voluntary participation of residents or pedestrians, are subject to staffing and budget constraints. As a result, the time span between assessments becomes extended [8]. Besides being user-oriented, the practicality of the proposed deep learning approach was also evident, as it achieved a high classification performance in detecting irregular walking surfaces using just a single wearable sensor. In contrast to the shallow machine learning approach, the proposed deep learning approach had an inherent ability to automatically distill knowledge from a high dimensional feature set to extract a good representation, which is analogous to automated feature selection, while simultaneously enhancing the classification performance. This scalable deep learning method could be deployed as a tool to continuously analyze pedestrians' acceleration data to monitor the surface conditions of sidewalks. The system, by removing human biases during the evaluation, could introduce new perspectives to walkability assessments and minimize the time and expense needed for on-site inspections.

Nonetheless, this study had its limitations. One of them was that the experiment was conducted in an experimental setting instead of real-world neighborhoods that vary in walkability. The subjects could have demonstrated distinct walking patterns in a real-world environment as opposed to the experimental setting. Another limitation was the small sample size of subjects. Therefore, we evaluated the approach by iterating through each

subject to draw conclusions repeatedly. Another drawback of the proposed framework was its inability to track the subjects' locations due to the lack of global positioning system (GPS) data. Finally, the LSTM is inherently a sequential model that relies on recurrence. This sequential nature impeded parallelization during training [55] and could present challenges in capturing long-term dependencies for longer data sequences in future experiments.

In the future, we plan to include more subjects to increase the number of subject samples and conduct our experiment in real-world walkable and less walkable neighborhoods. Furthermore, GPS data could be integrated in future works to locate participants and pinpoint the source of problematic sidewalk walking surfaces. To address the limitations of the LSTM and to further enhance the classification performance, we also plan to consider the use of a state-of-the-art deep network called the Transformer, which relies on a self-attention mechanism instead of recurrence.

## 6. Conclusions

In this study, we presented a novel classification approach for the automated detection of irregular walking surfaces based on a LSTM network using a single wearable accelerometer placed at the right ankle. Three input modalities for the training of the LSTM network were explored and their effectiveness was evaluated by comparing their classification performance to the traditional baseline support vector machine (SVM) machine learning method [10]. Based on the experimental results, it was found that the LSTM network trained with single-stride hand-crafted gait features with post-processing achieved the best performance. This affirmed the feasibility and effectiveness of the proposed approach. The results indicated that the proposed method could be used as an unbiased tool for detecting potentially problematic walking surfaces. Furthermore, this study unveiled new avenues for the development of real-time sidewalk assessment systems. Considering the sensitivity of our method to subtle variations in gait patterns, which is critical in clinical and therapeutic settings, the proposed method could potentially be extended to the healthcare domain as well. The application of the proposed method in monitoring gait patterns in individuals with neurological disorders or assessing the effectiveness of gait-improving interventions presents a promising direction for future research.

**Author Contributions:** Conceptualization: J.-H.Y., Y.N., X.Z. and H.R.N.; Methodology: X.Z., J.-H.Y. and H.R.N.; Formal analysis: H.R.N.; Investigation: H.R.N.; Resources: H.R.N.; Data Curation: J.-H.Y. and H.R.N.; Writing—original draft preparation: H.R.N.; Writing—review and editing: J.-H.Y., Y.N., X.Z. and H.R.N.; Supervision: J.-H.Y., X.Z. and Y.N. All authors have read and agreed to the published version of the manuscript.

**Funding:** This research received no external funding.

**Institutional Review Board Statement:** The study was conducted in accordance with the Declaration of Helsinki and approved by the institutional review board (or ethics committee) of the University of Nebraska Medical Center (protocol code 242-18-EP, approved 26 February 2020).

**Informed Consent Statement:** Informed consent was obtained from all subjects involved in the study.

**Data Availability Statement:** The data presented in this study are available on request from the corresponding author. The data are not publicly available because data has been collected for a specific research project, and the terms of data collection do not allow for public release.

**Conflicts of Interest:** The authors declare no conflict of interest.

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
