# Peer review of "Deep-Learning-Based Approach for Automated Detection of Irregular Walking Surfaces for Walkability Assessment with Wearable Sensor"

_applsci, doi:10.3390/app132413053_

Round 1

Reviewer 1 Report

Comments and Suggestions for Authors

-the paper makes a valuable contribution to the field of wearable technology and has the potential to improve sidewalk assessment practices and promote walkability. The proposed classification method is accurate and efficient, and it can be implemented using a single wearable accelerometer, making it a practical solution for real-world applications.

-Provide more details about the LSTM network architecture, including the number of layers, the number of neurons in each layer, and the activation function used. This would be helpful for researchers interested in implementing the method themselves.

-Discuss the potential impact of different preprocessing techniques on the performance of the LSTM network. For example, the authors could compare the performance of the proposed method with and without data normalization.

-Explore the potential applications of the proposed method beyond sidewalk assessment. For example, the authors could discuss how the method could be used to monitor gait patterns in individuals with neurological disorders or to assess the effectiveness of interventions aimed at improving gait.

- the paper is well-written and informative. However, the authors could enrich the paper by providing more details about the LSTM network architecture, discussing the potential impact of different preprocessing techniques, and exploring the potential applications of the proposed method beyond sidewalk assessment.

-Provide a table or diagram that summarizes the architecture of the LSTM network.

-Conduct a systematic review of existing methods for sidewalk assessment and compare their performance to the proposed method.

-Discuss how the proposed method could be used to monitor gait patterns in individuals with neurological disorders or to assess the effectiveness of interventions aimed at improving gait.

-The paper should be enriched with references to existing research papers on Deep Leaning Application of Big Data such as https://doi.org/10.58496/MJBD/2022/004. This would strengthen the argument and provide readers with additional resources for further exploration.

-To enhance the paper's impact, the author should discuss potential future directions for this research.

Reviewer 2 Report

Comments and Suggestions for Authors

The authors proposed a classification method for detecting irregular walking surfaces that uses the LSTM model to analyze gait data from a single wearable accelerometer. The work is good in the given domain. However, certain concepts need to be refined before they are recommended for publication. My first question is regarding the network. If LSTM is an established network, then what is the technical contribution of this work? Secondly, I did not find any feature analysis to feed the robust features to the network. There is no empirical analysis to validate the effectiveness of the employed feature descriptor experimentally. Therefore, it is recommended to include features analysis. Thirdly, please mention apparent potential drawbacks before contributions. For this, you can follow the paper entitled deep multiscale pyramidal features network for supervised video summarization and the article entitled deep learning based speech emotions recognition for parkinson patient. Fourthly, the authors are recommended to restructure the Related work. For this, you can follow the paper on an IoT enable anomaly detection system for smart city surveillance. Moreover, I am recommending concisely Writing the conclusion of your paper. 

Round 2

Reviewer 2 Report

Comments and Suggestions for Authors

There are no more comments from my side. I recommend the acceptance of the manuscript.